# Comprehensive Analysis of Kisspeptin Signaling: Effects on Cellular Dynamics in Cervical Cancer

**DOI:** 10.3390/biom14080923

**Published:** 2024-07-29

**Authors:** Deisy Yurley Rodríguez-Sarmiento, Paola Rondón-Villarreal, Pedro Henrique Scarpelli-Pereira, Michel Bouvier

**Affiliations:** 1Faculty of Health Sciences, Universidad Autónoma de Bucaramanga, Bucaramanga 680003, Colombia; 2Instituto de Investigación Masira, Facultad de Ciencias Médicas y de la Salud, Universidad de Santander, Bucaramanga 680003, Colombia; diseno.molecular@udes.edu.co; 3Department of Biochemistry, Institute for Research in Immunology and Cancer (IRIC), Université de Montreal, Montreal, QC H3T 1J4, Canada; pedro.henrique.scarpelli.pereira@umontreal.ca (P.H.S.-P.); michel.bouvier@umontreal.ca (M.B.)

**Keywords:** kisspeptin receptor, kisspeptin, cancer, cytotoxicity, signal transduction

## Abstract

Kisspeptin, a key neuropeptide derived from the *KISS1R* gene, is renowned for its critical role in regulating the hypothalamic–pituitary–gonadal axis and reproductive hormone secretion. Beyond its primary function in reproductive biology, emerging research has illuminated its influence in various cancers, mediating significant effects through its interaction with the G protein-coupled receptor, kisspeptin receptor. This interaction has been implicated in modulating cellular processes such as proliferation and metastasis, making it a potential target for therapeutic intervention. Our study initially screened ten kisspeptin-10 analogs through cytotoxic effects of kisspeptin-10 (KP10) and its analogs in several cancer types, including cervical, prostate, breast, and gastric cancers, with a particular focus on cervical cancer, where the most profound effects were observed. Further exploration using kinase array assays revealed that these analogs specifically alter key kinases involved in cancer progression. Migration assays demonstrated a substantial decrease in cell motility, and Bioluminescence Resonance Energy Transfer assays confirmed these analogs’ strong interactions with the kisspeptin receptor. Overall, our results indicate that these KP10 analogs not only hinder cervical cancer cell proliferation but also curtail migration through targeted modulation of kinase signaling, suggesting their potential as therapeutic agents in managing cervical cancer progression. This comprehensive approach underscores the therapeutic promise of exploiting kisspeptin signaling in cancer treatment strategies.

## 1. Introduction

G protein-coupled receptors (GPCRs) are critical for cellular signal transduction and are major targets in pharmaceutical research [1,2]. Structurally, GPCRs have N- and C-terminal regions and seven transmembrane segments [3]. Upon ligand binding, GPCRs activate G proteins, leading to downstream signaling like cAMP activation, ERK1/2 phosphorylation, and calcium mobilization [4].

Kisspeptin receptor is a G protein-coupled receptor (GPCR) that activates the G proteins Gα_q/11_ and its natural ligand is kisspeptin [5], a family of structurally related peptides derived from the *KISS1R* gene [6]. *KISS1R* encodes a precursor protein of 145 amino acids that is subsequently proteolytically cleaved to produce a series of C-terminal amidated peptides named kisspeptin-54 (KP54), kisspeptin-14 (KP14), kisspeptin-13 (KP13), and kisspeptin-10 (KP10), all of which have biological activity and are endogenous ligands for the kisspeptin receptor [7].

Currently, the KISS1/kisspeptin receptor system is known to be the main guardian of the reproductive axis in puberty and adulthood and plays a crucial role in the control of endocrine functions [7]. It serves as a critical regulator of gonadotropin-releasing hormone (GnRH) secretion, essential for puberty and fertility in both males and females. Kisspeptins stimulates the release of GnRH in the hypothalamus, which in turn triggers the secretion of gonadotropic hormones from the pituitary, thereby regulating reproductive function [8]. Beyond its reproductive role, the kisspeptin system has been shown to have significant non-canonical roles in various pathological conditions [9].

Recently, the kisspeptin system has garnered considerable attention in cancer research due to the associations between kisspeptins and their receptor with cancer progression and metastasis across various cancer types [10]. It has been documented that the kisspeptin receptor acts as a metastasis suppressor in prostate cancer, where low expression of the receptor is linked to increased tumor aggressiveness [11]. This suppressive behavior is consistent in gastric cancer, in which reduced receptor expression is also associated with tumor invasion and distant metastasis [12]. In contrast, in breast cancer, an upregulation of the *KISS1R* gene is observed, which is associated with more aggressive tumor phenotypes and an increased risk of mortality, indicating a potential oncogenic activity [13]. Thus, the kisspeptin system exhibits a complex and diverse role in oncology, predominantly acting as a metastasis regulator in prostate and gastric cancers, while it may exhibit oncogenic properties in breast cancer.

In the functional analysis of the kisspeptin system across different types of cancer, we can first observe that in prostate cancer, according to the study by Kim et al. [14], the activation of EIF2AK2 by kisspeptin suppresses tumor growth and metastasis. This suggests that kisspeptin-mediated signaling may play a crucial role in regulating the motility and invasion of prostate cancer cells. Additionally, in gastric cancer the suppressive effects of kisspeptin on tumor invasion are believed to be mediated through downregulation of matrix metalloproteinases (MMPs), particularly MMP-9 and MMP-2, which are critical in tumor invasion and metastasis formation [15]. In breast cancer, it has been described that KP10 can stimulate the migration and invasion of ERα-negative breast cancer cells via transactivation of the EGFR pathway and increased MMP-9 activity. These findings underscore the need for a nuanced understanding of kisspeptin/kisspeptin receptor signaling in breast cancer, as its role appears to be context-dependent and influenced by factors such as estrogen receptor status and interaction with other cellular pathways [16].

In cervical cancer, emerging studies have suggested that kisspeptin may play an inhibitory role in tumor advancement. Activation of the kisspeptin receptor by kisspeptin-10 has shown promising results in cytotoxicity assays [17]. Additionally, kisspeptin receptor has been identified as a potential tumor marker, as its expression is found to be almost 10 times higher in cervical tumor cells compared to non-tumor cells [18]. This is a crucial development considering that cancer ranks among the top causes of global mortality [19], and metastasis stands as a critical determinant of mortality in affected patients [20]. Further insights into the molecular mechanisms of cancer progression are provided by recent computational investigations, such different studies which explores potential inhibitory pathways that could be targeted by new therapeutic agents [21]. Another important parameter to consider is the complexity of redox modifications are crucial in cancer therapy because they can selectively disrupt cancer cell metabolism and signaling, potentially leading to targeted cell death while sparing healthy cells [22] and supports the need for a nuanced understanding of the biochemical pathways influenced by the kisspeptin system.

In this study, we analyzed the modifications in protein phosphorylation triggered by various kisspeptin analogs in cellular models of breast, prostate, cervical, and gastric cancers. Building on this, we specifically aimed to clarify the role of the kisspeptin system in cervical cancer by examining its influence on cellular signaling pathways and kinase phosphorylation dynamics using Bioluminescence Resonance Energy Transfer (BRET) biosensors and antibody arrays. This comprehensive approach revealed how kisspeptin significantly influences critical signaling pathways involved in the progression of various cancers, offering valuable insights. The emerging evidence underscores the potential of the kisspeptin system as a foundation for novel therapeutic strategies in oncology, highlighting its importance across diverse cancer types.

## 2. Materials and Methods

### 2.1. Peptide Synthesis

Eleven kisspeptin-10 analogs were synthesized by sequentially substituting each amino acid in the chain with alanine, as shown in Table 1. The kisspeptin-10 (KP10) analogs were synthesized using microwave-coupled solid-phase synthesis employing the Fmoc strategy [23]. The resin used was Rink Amide 0.40 mmol/g. The α-amino terminal protecting group for the amino acids was 9-fluorenylmethoxycarbonyl (Fmoc). Removal of this protecting group was carried out using 20% piperidine. The solvents utilized were N,N’-dimethylformamide (DMF) and methanol (CH_3_OH). Amino acid activation was achieved using N,N’-diisopropylcarbodiimide (DIC). Verification of coupling and deprotection was performed using 2,4,6-trinitrobenzenesulfonic acid (TNBS) and N,N-diisopropylethylamine (DIPEA). Once the desired sequence was obtained, the peptide was released from the solid support using trifluoroacetic acid (TFA). Finally, the peptides were purified by reverse phase solid phase extraction using Sep-Pak C18 microcolumns. The kisspeptin-10 (KP-10) peptide utilized in this study was purchased from China Peptides Co., Ltd. (Shanghai, China).

### 2.2. MALDI-TOF Mass Spectrometry

Peptide identification was performed on a Bruker ultraflextreme mass spectrometer with 40,000 FWHM resolution, 1 ppm accuracy, 337 nm laser, and a mass range of 500 to 3000 Da in the positive mode using SCiLs Lab software for Mass Spectrometry Imaging. A supersaturated solution of 2,5-dihydroxybenzoic acid (DHB) in ACN/H_2_O 30:70 with 0.1% TFA was used as matrix.

### 2.3. Cell Culture

Human cervical adenocarcinoma (HeLa) cells (cat# CRM-CCL-2) were obtained from the American Type Culture Collection (ATCC, Manassas, VA, USA). Human breast adenocarcinoma (MCF7) cells, Human prostate adenocarcinoma (PC3) cells, Human gastric adenocarcinoma (AGS) cells and Human Embryonic Kidney cells (HEK293T) were kindly provided by the Universidad Industrial de Santander. HeLa, MCF7, and HEK293T cells were cultured in DMEM medium supplemented with 10% FBS (fetal bovine serum), 1% penicillin/streptomycin. PC3 and AGS cells were cultured in RPMI medium supplemented with 10% FBS, 1% penicillin/streptomycin. All cells were maintained at 37 °C and 5% CO_2_.

### 2.4. Cytotoxicity MTT Assay

15,000 cells per well were plated in 96-well plates (Greiner, Fisher Scientific, Waltham, MA, USA) using a fully complemented medium, and incubated at 37 °C. Following a 24 h period, cells underwent serum starvation before being treated with KP10 and its analogs at concentrations of 10, 100, 250, and 500 nM for 48 h. Subsequently, each well received 100 µL of MTT solution (0.5 mg/mL), and the plates were incubated at 37 °C for 3 h. After adding the MTT solvent and further incubation for 15 min, the resultant absorbance was measured at 590 nm using a BioTech RT-2100C microplate reader. Data analysis was performed using Prism 8.0 software (GraphPad).

### 2.5. Phosphorylation in Human Phospho-Kinase Array

Cell lysates were prepared from cancer cells treated with 1 µM of KP10 and its analogs, as previously described [24]. These lysates were then incubated with arrays featuring immobilized phospho-specific antibodies [25,26]. To detail the treatment protocol: the cancer cells underwent a 16 h incubation in serum-free DMEM or RPMI as applicable, followed by a brief 10 min stimulation with KP10 and its analogs at 1 µM or with a vehicle as the control. This setup ensures the precise detection of phosphorylation events mediated by the treatment. The Human Phospho-Kinase Array (Human Phospho-Kinase Array, Proteome Profiler, R&D Systems, Minneapolis, MN, USA) allows parallel detection of the relative levels of phosphorylation of 43 kinase sites and two total proteins in cell lysates. Capture antibodies for each target were printed in duplicate on nitrocellulose membranes, which were blocked and incubated with diluted cell lysate overnight. After washing, a cocktail of biotinylated detection antibodies was added, followed by streptavidin-HRP. The addition of chemiluminescent reagents led to light emission at each spot, proportional to the amount of bound phosphorylated protein. The arrays were exposed to film, and the chemiluminescent signal was captured by BioRad Gel DocXR+ system (Bio-Rad Laboratories, Hercules, CA, USA) and the signal density was analyzed and processed using ImageJ software [27] to determine relative changes in kinase phosphorylation profiles between samples.

### 2.6. In Vitro Wound Healing Assay

Cells (3 × 10^5^ cells/well) were seeded in 24-well plates for 24 h. Then, a sterile pipette tip of 20–200 μL was held vertically to create a cross-shaped scratch in each well. The detached cells were carefully removed by washing with 500 μL PBS and shaking at 500 rpm for 5 min. Then, 500 μL of medium containing the peptides to be tested was added to each well and incubated for 72 h. Prior taking each image, the plate was washed with 500 μL of PBS and gently shaken for 30 s. Then, 500 μL of medium containing the corresponding peptides was added to each well again and images were taken at 24 h intervals using a Primovert microscope at 4× magnification and 1/3700 s exposure time. Wound opening areas were obtained using Prism 8.0 software (GraphPad).

### 2.7. BRET Assays

The Bioluminescence Resonance Energy Transfer (BRET) assay was developed as described before [28]. Briefly, HeLa cells were transiently transfected with these plasmids encoding for the GPCR, Gα subunit, the G protein effectors tagged with the BRET donor (RlucII), and the plasma membrane anchor (the CAAX motif from KRas, which includes a prenylation and a polybasic domain) fused to the BRET acceptor rGFP using Polietilendiimine (PEI) as described in Avet et al. [29]. Fourty-eight hours post-transfection, cells were incubated with the coelenterazine substrate for Renilla luciferase Prolume Purple, stimulated with kisspeptin receptor agonists for 10 min and BRET signals were measured using a dual-filter luminometer (SPARK, Tecan Life Sciences, Männedorf, Switzerland). BRET ratios were calculated by dividing the acceptor emission by the donor emission, providing a quantitative measure for the concentration of donor molecules on the plasma membrane. Data were analyzed using Prism 8.0 (GraphPad) to determine interaction dynamics under various experimental conditions.

### 2.8. Statistical Analyses

For the phosphorylation level assays, only those kinase substrates with intensity changes of ±30% that were statistically significant, as determined by a One-Way ANOVA test, were considered affected by the condition, as previously described by Xiao et al. [30]. The pathways activation in BRET assays and migration assay was analyzed using a One-Way ANOVA test, with statistical significance defined as *p* < 0.05. Data graphs and statistical analyses were generated and conducted using GraphPad software (GraphPad, San Diego, CA, USA).

### 2.9. Bias Factor Calculation

The bias factor for the kisspeptin analogs was calculated using the operational model of agonism as described by Kenakin [31]. Emax and EC50 values were determined for each signaling pathway (e.g., G protein activation, β-arrestin recruitment). The data were then fitted to the operational model to estimate transduction coefficients (τ). The bias factor (ΔLog(τ/K_A_)) was calculated by comparing the τ/K_A_ values for each pathway relative to a reference pathway. This method quantifies signaling bias by evaluating relative efficacy and potency across pathways. Statistical significance of the observed bias was assessed to ensure the differences were not due to random variation. All calculations and analyses were performed using GraphPad Prism software (GraphPad, San Diego, CA, USA).

### 2.10. Molecular Dynamics Simulations

The 3D structure of kisspeptin was predicted using the online tool PEP-FOLD3 [32] and the selected model was the one at the top of the best cluster. For the structure of kisspeptin receptor the prediction was performed using the online tool AlphaFold2 of ColabFold [33], and the quality of the best model was evaluated through the software tool Pro-SA-web [34]. Later, we used the Membrane Builder tool in the CHARMM-GUI [35] online server to build four systems, i.e., the kisspeptin receptor in its inactive form, the kisspeptin receptor together with the native kisspeptin, the kisspeptin receptor with the Ala^3^-KP10 peptide and the kisspeptin receptor with the Ala^4^-KP10 peptide. For the 4 systems, the kisspeptin receptor was immersed in a lipid bilayer composed by Cholesterol (CHL), Sphingomyelin (PSM), Phosphatidylcholine (POPC), Phosphatidylethanolamine (POPE), and Phosphatidylserine (POPS) according to the composition of a cancerous eukaryotic cell reported in the study of Klähn and Zacharias [36]. The system was surrounded by a rectangular simulation box (with periodic boundary conditions) of TIP3P water molecules and Ions of K+ and Cl− to neutralize the systems with an ion concentration of 0.15 M (physiological ion concentration).

First, an energy minimization of the systems was performed using the leap-frog algorithm for 5000 steps. Then, the systems were equilibrated under NPT conditions at 303.15 K and 1 bar pressure for 1500 ps, followed by 200 ns of production run under constant temperature and pressure. All the simulations were performed using GROMACS version 2024 [37].

## 3. Results

### 3.1. Design and MALDI-TOF Mass Spectrometry of KP10 Analogs

Earlier studies have shown that the kisspeptin-10 exhibits higher receptor affinity and evokes more potent functional responses than its extended counterparts like kisspeptin-13 and kisspeptin-14 [38]. To further explore the structural requirements of this core sequence, we have carried out Ala scan of human KP10 to identify important residues for kisspeptin receptor-agonistic activity. To this aim, we synthesized human KP10 Ala-analogs (Table 1) by standard Fmoc-based solid phase peptide synthesis, and for all synthesized peptides, the experimental molecular weight [M+H]+ observed by MS-MALDI-TOF analysis was in concurrence with the theoretical value calculated using the MS/MS Fragment Ion Calculator program (db system biology) [39] as shown in Table 1.

### 3.2. Effects of KP10 and Ala-Substituted Analogs on Cytotoxicity in Cancer Cells

The cytotoxic profiles of KP10 and its analogs across various cancer cell lines, MCF7, AGS, HeLa, and PC3, and HEK293T an immortalized cell line, were delineated, revealing notable variances in their biological activities. KP10 displayed differential sensitivity with the HeLa cell line being the most responsive, whereas AGS cells exhibited substantial resistance. Among the analogs, Ala^1^-KP10 emerged as consistently more potent across the panel, suggesting an enhanced interaction with the cellular targets. Notably, in HeLa cells, certain analogs, particularly Ala^3^-KP10 and Ala^4^-KP10, demonstrated heightened cytotoxic efficacy, as indicated by markedly low pIC50 values (Table 2). In the cytotoxicity assays, the analogs Ala^3^-KP10 and Ala^4^-KP10 demonstrated notable efficacy, particularly in HeLa cell lines. These findings highlight the potential of these analogs as therapeutic agents, with Ala^3^-KP10 and Ala^4^-KP10 showing significantly higher potency and effectiveness in inducing cell death compared to other tested analogs. The enhanced cytotoxic responses observed are attributed to their ability to disrupt key signaling pathways integral to cancer cell survival and proliferation. This specificity of response underscores potential selective affinities of the analogs for the molecular components unique to each cell type. These findings propel the rationale for further mechanistic studies to explore the structure-activity relationships governing the efficacy of KP10 analogs and their prospective therapeutic applicability in cancer treatment strategies.

### 3.3. Impact of Ala^3^-KP10 and Ala^4^-KP10 on Kinase Activation and Cell Migration Dynamics

The analogs Ala^3^-KP10 and Ala^4^-KP10 were selected for further analysis based on intriguing cytotoxicity results in cervical cancer cells (Table 2), which prompted a more detailed exploration of their effects on kinase phosphorylation and cell migration processes. Although their unique behavior in cervical cancer cells was particularly notable, these analogs were also assessed across other previously mentioned cell lines, including those derived from breast, prostate, gastric, and cervical cancers, to ensure a comprehensive understanding of their potential applications.

In the analysis of a kinase array (Human Phospho-Kinase Array, Proteome Profiler, R&D Systems, Minneapolis, MN, USA), when cervical cancer cells were stimulated with KP10 and its analogs, a distinct set of enzymes—Chk2, c-Jun, p53, p70 S6 kinase, RSK 1/2/3, STAT1, STAT3, STAT6, and PRAS40—emerged as key players (Figure 1).

Specifically, was observed a decrease in the phosphorylation levels of p53(S15) and PRAS40, suggesting a potential shift towards pathways less reliant on these tumor suppressors. Conversely, kinases such as Chk2, c-Jun, p70 S6 kinase, RSK 1/2/3, and members of the STAT family exhibited increased phosphorylation, indicating an enhanced pro-oncogenic signaling. These findings underscore the dualistic nature of kisspeptin signaling in modulating cancer cell behavior, highlighting its complex role in cervical cancer progression.

The impact of KP10 and its analogs on cellular migration was explored in all cancer cells through a wound healing assay, incorporating the analogs Ala^3^-KP10 and Ala^4^-KP10 as illustrated in Figure 2. In addition, the wound closure dynamics following treatment with KP10 and its analogs displayed intriguing variations in response across different cancer cell lines. HeLa and MCF7 cells exhibited moderate migration inhibition, with the most pronounced effects observed in cells treated with Ala^4^-KP10, particularly at the 48- and 72-h marks. This behavior stands in contrast to PC3 and AGS cell lines, where despite Ala^4^-KP10 remaining the more inhibitory treatment, the difference in the percentage of open wound area compared to the control was less marked. This pattern suggests that the response to KP10 and its analogs might be influenced by the intrinsic cellular characteristics specific to each cancer line, with HeLa and MCF7 showing a higher susceptibility to inhibition of cell migration. These findings highlight the need for further exploration of the underlying molecular interactions that govern these responses in the context of targeted cervical and breast cancer.

### 3.4. Analysis of KP10 and Relevant Analogs in Kisspeptin Receptor Signaling Transduction Pathways

Bioluminescence Resonance Energy Transfer (BRET) was employed to study the signaling profiles of the kisspeptin receptor upon stimulation with KP10 and its analogs Ala^3^-KP10 and Ala^4^-KP10. These signaling profiles were developed in HeLa cells, chosen for their relevance to cervical cancer based on results from cytotoxicity assays. This technique allowed for real-time monitoring of receptor activation and subsequent intracellular signaling events, providing a detailed understanding of the receptor dynamics and interactions triggered by these peptides. Given that KP10 is the endogenous ligand, the comparison of EC_50_ values across different G protein signaling pathways for Ala^3^-KP10 and Ala^4^-KP10 (Appendix A) offers an understanding of their pharmacological profile in the context of HeLa cells.

The endogenous ligand, KP10, exhibits the lowest EC_50_ values, indicating it has the highest potency and serves as a benchmark for efficacy in activating Gα_q_, Gα_11_, Gα_14_, and Gα_15_ pathways. This high affinity of the natural agonist highlights its critical role in physiological signaling processes and sets a standard for evaluating the potency of synthetic or external compounds. In contrast, Ala^3^-KP10, with EC_50_ values one order of magnitude higher than KP10, demonstrates reduced potency across all pathways. While capable of activating the signaling pathways, the increased EC_50_ suggests a lesser efficiency compared to the endogenous control. This indicates that while Ala^3^-KP10 can elicit responses through these G protein pathways, it may require higher concentrations to achieve effects comparable to the endogenous ligand, potentially affecting its therapeutic window and specificity. Ala^4^-KP10 exhibits the highest EC_50_ values, significantly surpassing both the natural ligand and Ala^3^-KP10, which denotes the lowest relative potency among the tested compounds. The substantial increase in EC_50_ values for Ala^4^-KP10, particularly in pathways mediated by Gα_11_ and Gα_15_, suggests markedly diminished efficacy in activating these signaling mechanisms. This reduced potency underscores the challenge in designing synthetic analogs that match the efficacy of endogenous ligands, emphasizing the need for further optimization to enhance their pharmacological activity. Also, the substitution of Trp with Ala and Asn with Ala in the KP10 peptide significantly reduces its potency in activating Gα_q_ protein-dependent signaling pathways, likely due to alterations in peptide-receptor interaction and conformational changes. The smaller, less polar alanine residues disrupt essential hydrophobic interactions and hydrogen bonding, crucial for the peptide proper folding and affinity towards its receptor. This modification impairs the peptide ability to induce necessary receptor conformational changes, leading to attenuated signal transduction. These findings underscore the critical role of specific amino acid residues in the structural and functional integrity of signaling peptides.

In the analysis of E_max_ values across Gα_q_ family signaling pathways, no significant differences were observed between the analogs and the KP10 (Appendix A). This suggests that while the analogs may vary in potency as indicated by their EC_50_ values, their maximum achievable effect (E_max_) in activating Gα_q_-mediated signaling remains comparable to that of the endogenous control (Figure 3). This uniformity in Emax underscores the analog’s ability to fully engage the receptor and elicit a maximal response, highlighting their functional similarity to the natural ligand in terms of the ceiling effect on Gα_q_ pathway activation.

Within the Gα_i/o_ family of G protein-coupled signaling pathways, widespread activation was not observed across the board, with the notable exception of the Gα_z_ protein pathway. Although Gα_z_ exhibits significantly stronger activation, it is important to note that some Gα_i_ proteins, such as Gα_i3_, may show minor activation. Additionally, the activation of proteins like Gα_oA_ and Gα_oB_ may be masked due to constitutive activity, as indicated by elevated basal levels. Thus, while the focus should primarily be on Gα_z_ due to its robust activation, it is noteworthy that other Gα_i/o_ proteins may still exhibit activation, underscoring the importance of comprehensive assessment in future investigations. (The activation curves of the Gα_i/o_ family can be found in the Appendix A Section of the manuscript). Here, differential EC_50_ values were recorded for KP10 and its analogs, Ala^3^-KP10 and Ala^4^-KP10, with KP10 showing an EC_50_ of 1.63 × 10^−7^ M, Ala^3^-KP10 with 6.46 × 10^−7^ M, and Ala^4^-KP10 with 1.10 × 10^−6^ M. Here, it can be seen that the substitution of Trp and Asn with Ala at positions 3 and 4 in KP10 crucially diminishes its potency in activating the Gα_z_ protein, reducing effectiveness approximately by a factor of ten. This alteration underscores the significant role of the Trp and Asn residues in facilitating the optimal interaction necessary for Gα_z_ pathway engagement. The considerable loss in activation potency due to these amino acid replacements highlights the specificity of KP10 interaction with Gα_z_, demonstrating how precise molecular configurations are essential for maintaining functional activity.

This distinct activation profile of the Gα_z_ pathway, in contrast to the lack of activation in other Gα_i/o_ family pathways, underscores the specificity of KP10 and its analogs interactions with G protein-coupled receptors. This selective activation of the Gα_z_ pathway by KP10 and its analogs, among the broader inactivity within the Gα_i/o_ protein family, highlights the unique pharmacological profile of these compounds. It raises intriguing questions about the role of Gα_z_-mediated signaling in the biological processes influenced by KP10 and its analogs. Further research into the downstream effects of Gα_z_ activation could unveil novel insights into the mechanisms by which KP10 and its analogs exert their effects, offering potential avenues for therapeutic intervention based on the modulation of this specific pathway.

The KP10 analogs also showed no activation of the Gα_s_ signaling pathway, as well as Gα_12_ and Gα_13_ proteins. This finding underscores the specificity of these compounds interactions with G protein-coupled receptors, highlighting their selective influence on certain pathways while leaving others, such as the Gα_s_, Gα_12_ and Gα_13_ pathways, unaffected. This selective activation pattern may have implications for understanding the biological roles of KP10 and its analogs and could guide the development of targeted therapeutic strategies.

Analyzing the EC_50_ values for β-arrestin1 and β-arrestin2 recruitment by KP10 and its analogs reveals insightful trends regarding their efficacy and selectivity in engaging these important signaling intermediates (Figure 3). Ala^3^-KP10, with EC_50_ values of 1.19 × 10^−6^ M for β-arrestin1 and 1.43 × 10^−6^ M for β-arrestin2, demonstrates a generally lower potency in recruiting both β-arrestins compared to KP10. The relatively close EC_50_ values for β-arrestin1 and β-arrestin2 suggest a more balanced interaction with both β-arrestins, albeit at a reduced efficacy. Ala^4^-KP10 presents the least potency in recruiting β-arrestins, with a markedly high EC_50_ value of 1.30 × 10^−5^ M for β-arrestin1 and 2.58 × 10^−6^ M for β-arrestin2. The significant disparity in the recruitment efficiency between β-arrestin1 and β-arrestin2, especially the pronounced reduction in β-arrestin1 recruitment, indicates a distinct signaling profile that could impact its biological and therapeutic utility.

To gain a clearer visualization of the signaling profiles across various analogs, we decided to calculate the bias factor [31] for Ala^3^-KP10 and Ala^4^-KP10 in different signaling pathways (Figure 4). Based on the provided Δ(logτ/KA) values for KP10 and its analogs, a comparative analysis reveals insights into their signaling bias across Gα_q_, Gα_11_, Gα_14_, Gα_15_ and Gα_z_ pathways. KP10, serving as the reference with a Δ(logτ/KA) of 0 across all pathways, establishes a baseline for evaluating the signaling bias of its analogs. In β-arrestin recruitment, the bias factor was not considered due to the very low values obtained, which did not yield usable data. The negative values observed for both Ala^3^-KP10 and Ala^4^-KP10 indicate a relative decrease in affinity or efficacy compared to KP10, reflecting a negative signaling bias. Notably, Ala^4^-KP10 exhibits a more pronounced negative bias than Ala^3^-KP10 across the board, except for Gα_14_, where Ala^3^-KP10 shows slightly greater bias. This trend is especially noticeable in Gα_z_ pathway, where Ala^4^-KP10 demonstrates a substantial decrease in preference (Δ(logτ/KA) of −1.66) compared to Ala^3^-KP10 Δ(logτ/KA) of −0.7. These findings suggest that specific modifications in KP10 analogs significantly impact their signaling through various G protein pathways relative to native KP10. The pronounced decrease in Ala^4^-KP10 affinity or efficacy, particularly towards the Gα_z_ pathway, warrants further investigation into the molecular interactions or receptor-ligand conformations driving these differences. In summary, KP10 analogs exhibit distinct signaling bias profiles compared to KP10, with Ala^4^-KP10 showing a more marked tendency towards negative bias in the activation of examined signaling pathways, providing an intriguing foundation for future research on peptide modifications in GPCR signaling modulation and therapeutic potential.

On the other hand, each transmembrane of the kisspeptin receptor reported in the study of Kotani et al. [40] was analyzed to check their possible structural changes between the inactive and the active form of the receptor, and also the possible structural changes when Ala^3^-KP10 and Ala^4^-KP10 peptides interact with the kisspeptin receptor (see Figure 5). These systems were chosen based on preliminary experimental data indicating significant interactions and notable effects on receptor behavior. While other analogs were synthesized and evaluated in different assays, the focus of these simulations was to provide a detailed mechanistic understanding of the most promising candidates. For these analyses, molecular dynamics simulations were conducted and the final structures (after 200 ns) for each transmembrane in each simulation were extracted and RMSD values were calculated between them (see Table 3). Notably, significant conformational changes were observed only in the TM4 region of the receptor-Ala^4^-KP10 complex. These findings suggest that the substitution of tryptophan residues at position 4 with alanine in the KP10 sequence induces unique structural rearrangements within the receptor’s TM4 region, which could have implications for the receptor’s activation and signaling mechanisms.

## 4. Discussion

This study on the pharmacological properties of KP10 and its analogs illuminates the contextual variability in the efficacy of kisspeptins across different cancer types, revealing metastasis-suppressing effects in various models but also contradictory roles, such as in hepatocellular carcinoma [41]. This contradiction highlights the diversity of intracellular networks triggered by kisspeptin signaling, involving pathways like MAPK, Akt, Wnt/β-catenin, and CXCR4, which regulate proliferation, survival, angiogenesis, and other pro-metastatic behaviors. Our findings indicate that activation of the kisspeptin system leads to a reduction in the phosphorylation of p53(S15) and PRAS40, suggesting an alternative signaling mechanism in cancer biology that might prioritize cell survival over apoptosis. This demonstrates that kisspeptin activation modulates tumor progression pathways by reducing the activity of the tumor suppressor p53 and PRAS40, which are crucial for inhibiting tumor growth and promoting cell death. Furthermore, an observed increase in the phosphorylation of kinases such as Chk2, c-Jun, p70 S6 kinase, RSK 1/2/3, and members of the STAT family upon kisspeptin system activation highlights its multifaceted role in cancer biology. This elevation in kinase activity indicates a potential pro-oncogenic influence of kisspeptin signaling, promoting pathways associated with cell proliferation, survival, and metastasis, and underscores the need for targeted research to fully understand its therapeutic implications and risks in cancer treatment.

The interaction between Chk2, c-Jun, p53, p70 S6 kinase, RSK 1/2/3, STAT1, STAT3, STAT6, and PRAS40 underlines a complex signaling network in cancer biology. p53 and Chk2 emerge as guardians against oncogenesis, with p53 halting cell proliferation through apoptosis and cell cycle arrest, while Chk2 enhances this defense through both p53-dependent and independent pathways to counter DNA damage [42]. Conversely, c-Jun promotes cellular proliferation and oncogenic progression, regulated by AP-1 pathway activation [43]. Within the mTOR pathway, p70 S6 kinase and PRAS40 act as key modulators of cancer progression, influenced by their phosphorylation status [44]. RSK supports this network by promoting growth and survival downstream of the Ras-MAPK pathway [45], while the STAT family, particularly STAT3 and STAT5, play a crucial role in cell proliferation and immune evasion, marking them as potential therapeutic targets [46]. Together, these elements comprise a complex signaling mosaic, offering multiple targets for cancer therapy.

In concordance with the observed inhibitory effects on HeLa cell migration by KP10 and its analogs, an inverse relationship was identified between cellular motility and the phosphorylation status of STAT1, STAT3, and STAT6. Notably, treatment with KP10 analogs resulted in heightened levels of phosphorylated STAT1 which is commonly associated with anti-proliferative and anti-migratory responses [47], potentially elucidating the mechanism behind the reduced migration rates. Conversely, although STAT3 phosphorylation was also observed, its expected pro-migratory action was not realized, suggesting a non-canonical role of STAT3 in this context or the predominance of STAT1 and STAT6 signaling pathways in mediating the cellular response to these treatments. These findings indicate a complex interplay between STAT family members in governing cell migration, where the activation of STAT1 and STAT6 may override the promigratory signals of phosphorylated STAT3, thereby aligning with the overall diminished migratory capacity of HeLa cells upon exposure to KP10 analogs. Also, treatment with both analogs led to a significant increase in the phosphorylation of key regulatory proteins Chk-2, p70 S6 kinase, cJun, and RSK. The elevated phosphorylation levels of Chk-2 and p70 S6 kinase are indicative of enhanced DNA damage response and protein synthesis regulation [48], potentially contributing to cellular stabilization and reduced migratory activity. Concurrently, the activation of cJun and RSK, through phosphorylation, may suppress migration by modulating gene expression related to cellular adhesion and cytoskeletal dynamics [45,49]. Collectively, these phosphorylation events present a molecular signature that correlates with the observed inhibition of cellular migration, suggesting a multifaceted inhibitory mechanism exerted by the KP10 analogs on the migratory capacity of cancer cells.

This study also delves into the efficacy of KP10 and its analogs across diverse G protein signaling pathways, with a focus on cervical cancer cells, to elucidate their pharmacological profiles and potential therapeutic implications. The determination of EC_50_ values for KP10 and its analogs across pathways such as Gα_q_, Gα_11_, Gα_14_, Gα_15_, and notably, the Gα_z_ pathway, along with their recruitment of β-arrestin1 and β-arrestin2, provides a foundational understanding of their interactions with the kisspeptin receptor (Figure 3). KP10 is shown to possess remarkable potency in activating the Gα_q_ pathway, highlighting its significant role in modulating cellular processes pertinent to cancer progression. This potent activation suggests the influence of KP10 on critical pathways involved in cell proliferation, survival, and potentially metastasis [15].

The analysis extends to the comparison of KP10 analogs, revealing a reduced ability to activate the Gα_q_ pathway, suggesting a diminished impact on tumorigenesis and metastasis signaling cascades in cervical cancer. Furthermore, KP10 selective activation of the Gα_z_ pathway, with significantly lower EC_50_ values compared to its analogs, underscores a unique interaction with kisspeptin receptor, highlighting the need for further exploration into Gα_z_ role in cancer biology (Figure 3). This study also examines the differential recruitment of β-arrestins by KP10 and its analogs, indicating a potential preference for β-arrestin2-mediated pathways by KP10. This preference could implicate receptor desensitization, internalization, and the initiation of distinct downstream signaling events, shedding light on the complex balance of activating versus inhibitory signals that dictate cellular outcomes in their presence (Figure 3).

The differential responses observed in cervical and breast cancer compared to gastric and prostate cancers may reflect the physiological role of kisspeptin in the reproductive system, suggesting a nuanced influence based on sex-determined factors. Given kisspeptin’s integral role in reproductive hormone regulation, it is conceivable that its signaling pathways might exert antitumoral effects in female-associated cancers such as cervical and breast, by influencing pathways that restrict proliferation and metastasis. Conversely, in male-associated cancers such as prostate cancer, the same signaling pathways could potentially have protumoral effects, perhaps by aligning with the mechanisms that promote cell growth and survival. This hypothesis underscores the complex interplay between the physiological roles of peptides and their influence on cancer pathophysiology, which can differ markedly between sexes and cancer types. Further studies are warranted to elucidate the sex-specific oncological roles of kisspeptin and its potential as a differential target in cancer therapy.

## 5. Conclusions

The comprehensive analysis presented in this study not only underscores the therapeutic potential of KP10 and its analogs in cervical cancer cells but also emphasizes the complex modulation of cancer-related signaling pathways by these peptides. The nuanced understanding of kisspeptin receptor-mediated signaling, increased by the study insights into the context-specific modulation of cancer cell signaling by Ala^3^-KP10 and Ala^4^-KP10, lays the groundwork for future development of diagnostics or therapeutics. As such, the elucidation of the intricate molecular intersections between kisspeptins and oncogenic pathways becomes essential for advancing our understanding and treatment of cancer, highlighting the complexity and potential of targeting these pathways in cancer therapy.

## Figures and Tables

**Figure 1 biomolecules-14-00923-f001:**
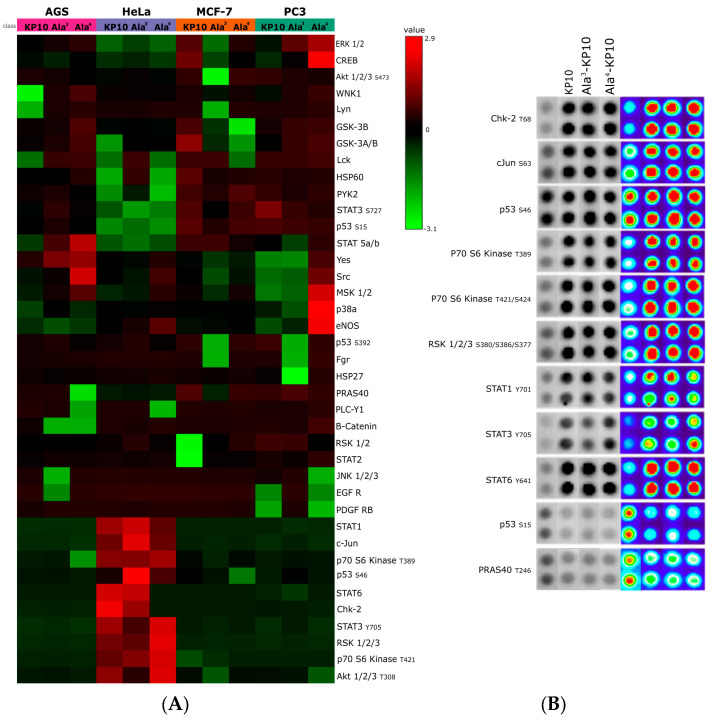
Kinase Activity Profiles Across Cell Lines. (**A**) This heatmap displays the kinase activity profiles in AGS, HeLa, MCF7, and PC3 cell lines following stimulation with KP10 and its analogs Ala^3^-KP10 and Ala^4^-KP10 1 μM for 10 min. Each row represents a different kinase, while columns correspond to the various treatment conditions across the four cell lines. Color intensity indicates the level of kinase activation or inhibition, providing a comparative visualization of the cellular responses to KP10 and its analogs. (**B**) Examination of phosphorylation levels in 11 human kinase substrates, which exhibited changes in their phosphorylation status following stimulation with KP10, Ala^3^-KP10, or Ala^4^-KP10 in HeLa cells. The analysis captures the phosphorylation dynamics of these substrates, where spot intensities have been transformed into a color-coded scale to underscore the observed variations. Original Western blot images are available in Appendix A.

**Figure 2 biomolecules-14-00923-f002:**
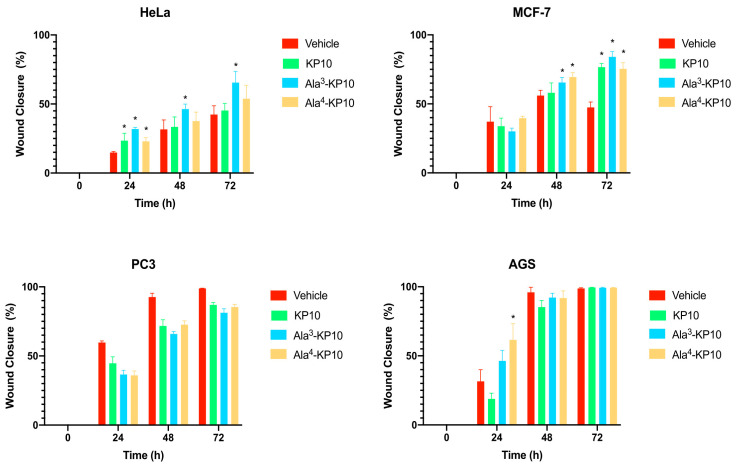
Temporal evolution of wound closure in HeLa, MCF7, PC3 and AGS cells subjected to KP10, Ala^3^-KP10, and Ala^4^-KP10. The graph displays wound healing assay results showing the percentage of wound closure (% migration) for different kisspeptin analogs. The data are presented as the percentage of the initial wound area that has closed, providing a clearer representation of cell migration. This combined visual analysis demonstrates the distinct impact of each peptide on the healing process, providing a clear comparison of their effectiveness at various time intervals. (* *p* < 0.05 versus vehicle).

**Figure 3 biomolecules-14-00923-f003:**
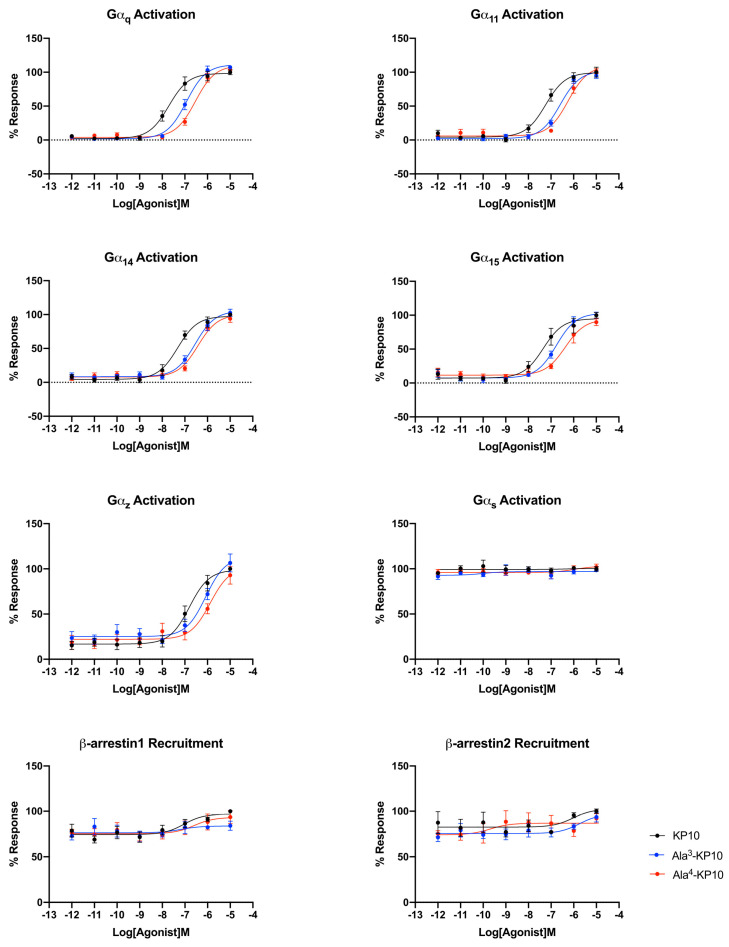
Comparative activation of the kisspeptin receptor through different signaling pathways in cervical cancer cells by KP10, Ala^3^-KP10, and Ala^4^-KP10. The graphs illustrate the Emax values for each pathway, indicating the maximal response achieved. Specific pathways analyzed include Gα_q_, Gα_11_, Gα_14_, Gα_15_, Gα_z_ Gα_s_ and β-arrestin1 and β-arrestin2 pathways, demonstrating the differential activation profiles of the analogs. Data were generated from four independent experiments.

**Figure 4 biomolecules-14-00923-f004:**
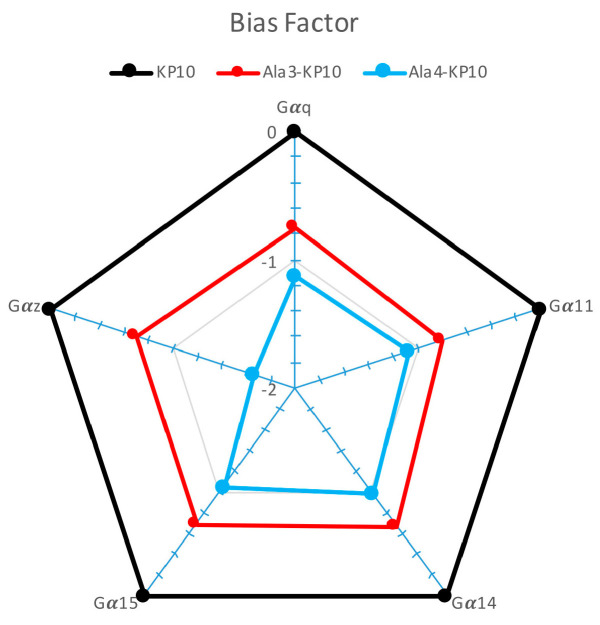
Bias factor of KP10 and analogs across Gα_q_, Gα_11_, Gα_14_, Gα_15_, and Gα_z_ Pathways. This plot presents the bias factor of KP10 and its analogs Ala^3^-KP10 and Ala^4^-KP10 toward activating Gα_q_, Gα_11_, Gα_14_, Gα_15_, and Gα_z_ signaling pathways in cervical cancer cells. ∆Log(τ/KA) ratios for Ala^3^-KP10 and Ala^4^-KP10 were plotted in “radial graphs” using a logarithmic scale to represent normalized transduction coefficients obtained in each pathway.

**Figure 5 biomolecules-14-00923-f005:**
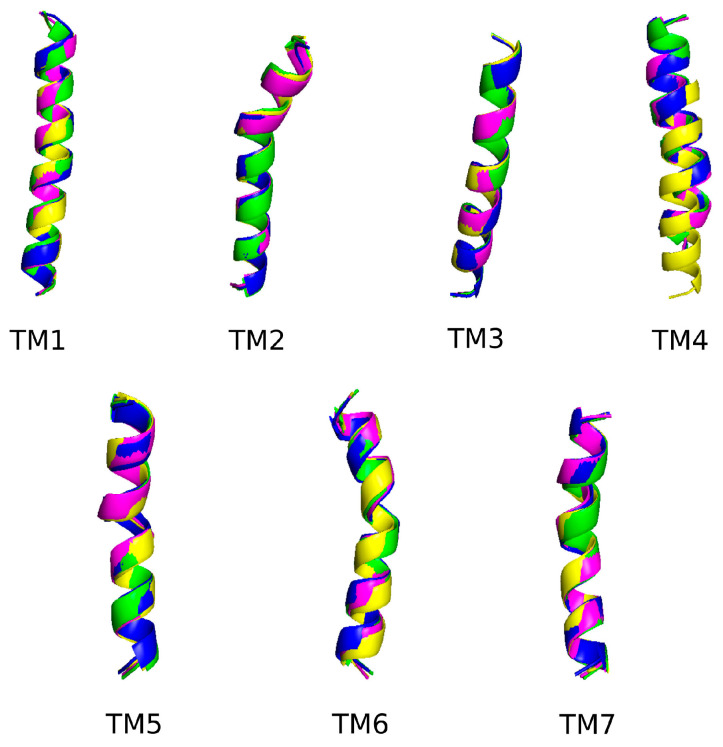
Comparative of the final structures for each transmembrane in the four systems, i.e., inactive form (magenta), with KP10 (green), with Ala^3^-KP10 (blue), and with Ala^4^-KP10 (yellow).

**Table 1 biomolecules-14-00923-t001:** Amino acid sequences and Mass Spectrometry values of synthesized KP10 analogs. This table lists the amino acid sequences of peptides utilized in our research, detailing their composition and modifications. Calculated indicates the theoretical monoisotopic MH+ value. Observed values were assessed by MALDI-TOF in the reflector mode unless otherwise noted.

Peptide	Sequence										Mass Spectrometry (Da)	
											Calculated	Observed
**KP-10**	**Tyr**	**Asn**	**Trp**	**Asn**	**Ser**	**Phe**	**Gly**	**Leu**	**Arg**	Phe	1304	1303.537
Ala^1^-KP10	Ala	Asn	Trp	Asn	Ser	Phe	Gly	Leu	Arg	Phe	1211	1211.234
Ala^2^-KP10	Tyr	Ala	Trp	Asn	Ser	Phe	Gly	Leu	Arg	Phe	1261	1261.461
Ala^3^-KP10	Tyr	Asn	Ala	Asn	Ser	Phe	Gly	Leu	Arg	Phe	1188	1188.489
Ala^4^-KP10	Tyr	Asn	Trp	Ala	Ser	Phe	Gly	Leu	Arg	Phe	1260	1260.589
Ala^5^-KP10	Tyr	Asn	Trp	Asn	Ala	Phe	Gly	Leu	Arg	Phe	1288	1288.241
Ala^6^-KP10	Tyr	Asn	Trp	Asn	Ser	Ala	Gly	Leu	Arg	Phe	1228	1228.284
Ala^7^-KP10	Tyr	Asn	Trp	Asn	Ser	Phe	Ala	Leu	Arg	Phe	1318	1318.354
Ala^8^-KP10	Tyr	Asn	Trp	Asn	Ser	Phe	Gly	Ala	Arg	Phe	1262	1262.203
Ala^9^-KP10	Tyr	Asn	Trp	Asn	Ser	Phe	Gly	Leu	Ala	Phe	1218	1218.613
Ala^10^-KP10	Tyr	Asn	Trp	Asn	Ser	Phe	Gly	Leu	Arg	Ala	1227	1226.697

**Table 2 biomolecules-14-00923-t002:** pIC50 values from cytotoxicity assays across cell lines. This table compiles the pIC50 values derived from cytotoxicity assays performed with KP10 and its analogs on several cancer cell lines, MCF7, AGS, HeLa, and PC3 and including HEK293T an immortalized cell line. Each cell line response to the treatment is quantified to reflect the inverse logarithmic concentration of the compound required to inhibit cell viability by 50% (pIC50), providing a comparative measure of the analogs’ cytotoxic potency.

Peptide	pIC50				
	HEK293T	HeLa	MCF7	PC3	AGS
KP-10	2.489	3.209	3.106	2.678	43.88
Ala^1^-KP10	2.311	2.483	2.947	2.475	50.33
Ala^2^-KP10	2.918	0.5728	2.756	2.67	38.71
Ala^3^-KP10	2.959	−0.6081	2.624	2.737	51.7
Ala^4^-KP10	3.489	−0.2705	3.132	2.541	43.25
Ala^5^-KP10	3.335	2.61	3.327	2.203	43.42
Ala^6^-KP10	3.518	2.523	3.168	2.461	38.05
Ala^7^-KP10	3.027	2.554	3.009	2.42	49.16
Ala^8^-KP10	4.017	2.564	3.15	2.151	34.5
Ala^9^-KP10	4.197	2.429	3.183	2.295	36.86
Ala^10^-KP10	3.292	2.493	2.826	1.739	34.56

**Table 3 biomolecules-14-00923-t003:** RMSD (Å) distances for each transmembrane of the kisspeptin receptor in its inactive form in comparison with the active form with the kisspeptin, with Ala^3^-KP10 peptide, and with Ala^4^-KP10 peptide.

Inactive	KP10	Ala^3^-KP10	Ala^4^-KP10
TM1	0.868	0.937	0.529
TM2	1.254	0.510	0.803
TM3	0.525	0.665	0.495
TM4	0.711	0.568	1.880
TM5	0.770	0.849	0.604
TM6	1.046	1.096	1.118
TM7	1.045	0.751	0.941

## Data Availability

The data presented in this study are available on request from the corresponding author.

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
