# Peer review of "Comprehensive Analysis of Kisspeptin Signaling: Effects on Cellular Dynamics in Cervical Cancer"

_biomolecules, 2024, doi:10.3390/biom14080923_

Round 1

Reviewer 1 Report

Comments and Suggestions for Authors

June 1, 2024

Ms. Ref. No.: biomolecules-3040384

Journal: Biomolecules.

Title: Comprehensive analysis of kisspeptin signaling: effects on cellular dynamics in cervical cancer.

Comments:

Thank you for your efforts in composing an on such a pertinent subject. I have taken the liberty of providing you with a few observations that I believe will serve to enhance the quality of your work. Please find my feedback outlined in the following paragraphs

1-      What was the main source of 10 synthesized analogs of Kisspeptin? Purchased or produced in laboratories where done this study?

2-      There are some  different cell line in this manuscript, (HeLa) cells , Human breast adenocarcinoma 96

(MCF7) cells, Human prostate adenocarcinoma (PC3) cells, Human gastric adenocarcinoma (AGS) cells and Human Embryonic Kidney cells (HEK293T), why these different lines? And additionally, what about non cancer cell line?

3-      The cytotoxicity MTT Assay was done for 24-hour periods, why was not followed for 48-hour periods?

4-      How long was the incubation in Phosphorylation in Human Phospho-Kinase Array?

5-      According to figure 3 , Why BRET Assays was done only for HeLa cells?

6-      The Molecular dynamics simulations was designed for four systems, involves the kisspeptin receptor in its inactive form, the kisspeptin receptor together with the native kisspeptin, the kisspeptin receptor with the Ala3-KP10 peptide and the kisspeptin receptor with the Ala4-KP10 peptide, what about other interaction with different analogs?

7-      According to table 2, how can introduce the cytotoxic efficacy of Ala3-KP10 and Ala4-KP10 in HeLa cells between another analogs? How can clarify the high value of AGS?

8-      In order to improve the clarity of the introduction, it is recommended that you include some of the following sources as references:

·         https://doi.org/10.3390/biom12111612

·         https://doi.org/10.1007/s00210-023-02551-0

9-       It seems to be better that recheck the spell of Citotoxicity in line 103.

Reviewer 2 Report

Comments and Suggestions for Authors

This manuscript tackles a complicated signaling system, further complicated by differential effects in different cancer types. The authors suggest that the Kisspeptin (KP) – Kisspeptin Receptor (KR) pathway could be a target in cancer treatment. However, the manuscript writing is not easy to follow in places, and the logic of the flow of experiments is not clear. The background does not clearly lay out the possible dual role of KP – KR as a potential tumor suppressor pathway or potential oncogenic pathway. Clearer presentation of past data on expression levels of both KP and KR (i.e. up or down regulated) in the different cancer types is needed to set the current study into context. As well as reporting any functional studies done. Overall, it is difficult to follow through the results sections as to whether the authors are saying this signaling pathway is acting as a tumor suppressor pathway or as an oncogenic pathway. More clear writing is needed to facilitate interpretation of the data and of the authors’ interpretations of the data.  

In the methods, there is no description of statistical methods used to analyze the data, and in the results, there is no indication that statistics were performed, even where the word significant was used.  This is a serious flaw in this manuscript.  

Also in the methods section, under the peptide synthesis section, there is no description of the Ala-scan method with the paragraph, though it is mentioned in the results section. 

It is unclear why different concentrations of peptides were used in the cytotoxicity assays versus the phosphorylation assays. Why was 1 uM chosen for the phosphorylation assays? Also, was the treatment time in these assays 48 h as it was in the cytotoxicity assays? This is not stated in the figure legend or in the methods. The figure legend should state both the treatment concentration and time length. 

Figure 5 should be within the results section, not in the discussion section. Moreover, it is not even mentioned in the text of the manuscript.  

Minor concerns: 

  • -- In several places in the text and figure / table legends, HEK293T cells are included in a list of ‘cancer’ cells, but while these cells are immortalized, they are not cancer cells. 

  • -- The units in Table 2 are not clear to someone not familiar with pIC50 values. Please state in table information.  

  • -- The figure 3 legend needs to be more clear on what is being shown, i.e. Emax. 

  • -- In figure 4, the data may be more easily interpreted, in the context of cancer analysis, as % wound closure which would be % migration, rather than % open wound area. 

Comments on the Quality of English Language

The English language is fairly good, and only minor editing is needed.

Round 2

Reviewer 1 Report

Comments and Suggestions for Authors

Thanks 

Author Response

Thank you for your time and dedication in reviewing our manuscript. Your insightful comments and suggestions have been invaluable in helping us improve the quality and clarity of our work. We greatly appreciate your efforts and thoughtful feedback.

Reviewer 2 Report

Comments and Suggestions for Authors

The authors have made significant changes to the manuscript, enhancing it’s quality. However, there is still some additional editing needed for clarity.

1.  In the background, in the first paragraph of the revised section, the authors state the following:

“Currently, the kisspeptin system has gained considerable attention in cancer research, as kisspeptins and their receptor have been linked to cancer progression and metastasis across various types of cancer [10]. Studies have highlighted the kisspeptin receptor's role as a metastasis suppressor in prostate cancer [11], contrasting with its impact in gastric cancer, where low kisspeptin receptor expression is linked to tumor invasion and distant metastasis [12]. In breast cancer, KISS1R gene upregulation is associated with aggressive tumor phenotypes and increased mortality risk [13]. In summary, the kisspeptin system is quite unique and it has not been possible to generalize its behavior in cancer thus far. The literature shows that in prostate cancer, the kisspeptin system acts as an inhibitor of metastatic processes, whereas in gastric and breast cancers, it increases aggressiveness. This is based on studies presenting the corresponding expression levels of the genes within the kisspeptin system [11-13].”

In cancer, upregulation is suggestive of an oncogenic activity, while down regulation is suggestive of tumor suppressor activity, although there are some exceptions to this, such as p53 appears to be upregulated, but it is a mutated / inactive form.  In gastric cancer, you report the behavior as contrasting with prostate cancer; however, it is not. Low kisspeptin receptor would indicate a tumor suppressor function of the pathway, not an oncogenic function. So, prostate and gastric cancer are consistent with tumor suppressor function. Breast cancer is the outlier, where it seems to have oncogenic activity due to the upregulation.  In the second paragraph of the revised section, the description is consistent with including the signaling in gastric cancer as tumor suppressive activity.

In the paragraph starting on line 79, you mention cervical cancer in the first sentence, but no detail is provided as to what the evidence is from the reference that makes it fall into the tumor suppressor pathway. Since cervical cancer is emphasized in the abstract, more detailed information is needed. At least one sentence needs to be added on the what the data in the reference shows, i.e. how do they show tumor suppressor activity?

2. In the graphs in Figure 2, the Y axis is now listed as Wound Closure (%), but the graphs are exactly the same as in the first version of the manuscript. So, does this mean that the graphs were mislabeled in the first version? Also, no statistical analysis is indicated in the graphs. Are the decreased migration activities seen in the graph, particularly with the breast,  prostate and gastric cancer cells lines significant? Asterics should be used in the graphs to indicate which analogs and doses are statistically significantly different from control.

3.  If the graphs in Figure 2 are actually presented in the revised manuscript as % wound closure, i.e. migration, then cervical cancer was the least responsive to KP10 or its analogs. In the current text, it is unclear why it was chosen for the comparative activation assay in Figure 3. In the text, the authors need to indicate the cytotoxicity assays as the rationale for using HeLa cells in Fig. 3. This should be added somewhere around lines 322 to 325.
